# NAP Family CG5017 Chaperone Pleiotropically Regulates Human AHR Target Genes Expression in *Drosophila* Testis

**DOI:** 10.3390/ijms20010118

**Published:** 2018-12-29

**Authors:** Angelina A. Akishina, Julia E. Vorontsova, Roman O. Cherezov, Mikhail S. Slezinger, Olga B. Simonova, Boris A. Kuzin

**Affiliations:** Koltzov Institute of Developmental Biology, Russian Academy of Sciences, Vavilova str. 26, Moscow 119991, Russia; ilitiri@bk.ru (A.A.A.); vjul83@mail.ru (J.E.V.); ro-tcherezov@yandex.ru (R.O.C.); milari2016@mail.ru (M.S.S.); kuzinb@yandex.ru (B.A.K.)

**Keywords:** nucleosome assembly protein, Aryl hydrocarbon receptor, xenobiotic, spermatogenesis, *CG5017*, *Drosophila*

## Abstract

To study the regulatory mechanism of the Aryl hydrocarbon receptor (AHR), target genes of transcription are necessary for understanding the normal developmental and pathological processes. Here, we examined the effects of human AHR ligands on male fecundity. To induce ectopic human *AhR* gene expression, we used *Drosophila*
*melanogaster* transformed with human *AhR* under the control of a yeast *UAS* promoter element capable of activation in the two-component *UAS-GAL4* system. We found that exogenous AHR ligands decrease the number of *Drosophila* gonadal Tj-positive cells. We also found both an increase and decrease of AHR target gene expression, including in genes that control homeostasis and testis development. This suggests that gonadal AHR activation may affect the expression of gene networks that control sperm production and could be critical for fertility not just in *Drosophila* but also in humans. Finally, we found that the activation of the expression for some AHR target genes depends on the expression of testis-specific chaperone CG5017 in gonadal cells. Since CG5017 belongs to the nucleosome assembly protein (NAP) family and may participate in epigenetic regulation, we propose that this nucleotropic chaperone is essential to provide the human AHR with access to only the defined set of its target genes during spermatogenesis.

## 1. Introduction

There is evidence that the Aryl hydrocarbon receptor (AHR) plays an important role in normal development and cancerogenesis [1,2,3,4,5,6,7,8,9,10,11,12,13,14]. A proper concentration of activated AHR is important for cell survival and organism functioning [15,16,17,18,19]. Ligand binding is critical for AHR activation, because after the binding it moves to the nucleus, dimerizes with the Aryl hydrocarbon receptor nuclear translocator (ARNT), and starts functioning as a transcriptional factor which binds to specific DNA sequences known as the xenobiotic response elements (XRE), driving expression of its target genes [20,21]. An increased risk of cancer and the inability to protect cells against the toxic effects of xenobiotics are the most dramatic consequences of the decreased AHR expression [22,23]. Activation of AHR in inappropriate tissues and organs causes abnormal development, including disorders in the immune, nervous, endocrine, cardiovascular, and generative systems. Among AHR target genes, there are many genes that encode proteins responsible for homeostasis, and maintaining these is necessary for successful adaptation in new ecological niches [21,24]. Human and mammal AHRs are activated by different endogenous and exogenous ligands (xenobiotics), while in invertebrates, AHR is activated only by endogenous ligands [3,24,25,26,27,28,29]. There is a wide range of affinities of xenobiotic ligands to AHR [30]. It is believed that the ligand binding affinities can modulate AHR’s ability to trigger the expression of its target genes [31]. The growth of the chemical and pharmaceutical industries has created conditions under which every person has a risk of exposure to xenobiotics. This may result in a variety of activations of ectopic AHR target genes in different tissues and at different stages of development.

Cell culture experiments do not provide a full understanding of the effects induced by AHR ectopic expression on the development of living organisms. For a better understanding of the function of human AHR in vivo, we created “humanized” *Drosophila* transgenic flies, which carry transgenes with the controlled expression of the human *AhR* gene guided by the yeast upstream activation sequence (*UAS*) [32]. This transgenic construct permits the induction of human AHR expression in certain *Drosophila* organs and cells with the help of tissue-specific *GAL4*-driver lines expressing a yeast GAL4 activator capable of recognizing *UAS* sequences and driving the transcription of downstream genes [33]. As most human AHR exogenous ligands are not capable of activating the *Drosophila* AHR homolog, this allows us to estimate the specificity of their action when adding them into the *Drosophila* food medium. It was shown that mouse AHR and *Drosophila* ARNT homolog (Tango) could form a functional transcriptional complex capable of inducing the dioxin-mediated activation of AHR target gene expressions in *Drosophila* [28]. 

We decided to use *Drosophila* as a verified model system to investigate the in vivo effects of human AHR ligands (xenobiotics) during development. In previous experiments using *UAS-AhR/GAL4-driver* flies, we have demonstrated that AHR activation could both increase and decrease transcription of AHR target genes in different tissues, and found that this effect depends on the developmental stage of the animal [32]. It is important to note that the effect of xenobiotics on the levels of AHR target gene activity was more clearly manifested in organs with a high number of proliferating cells. In adult organs, in which cell proliferation is complete, the actions of ligands on AHR target gene transcription levels were not detected. We found that the ligand’s effect on AHR target gene expression is mediated by the polycomb group (PcG) epigenetic chromatin regulators [32]. A similar xenobiotic effect may be expected in humans. In other words, we may expect the absence of the xenobiotic’s action on differentiated cells in humans. However, in adult humans, like in *Drosophila* imagoes, there are organs in which cells are continuously dividing. Some of these organs are testes. It is expected that ectopic AHR expression induced by xenobiotics may be the cause of various disturbances of spermatogenesis in humans. 

In this paper, we apply *Drosophila* transformed with human *AhR* to estimate a possible negative effect induced by xenobiotics on human spermatogenesis. We demonstrated that the ectopic activation of human AHR in *Drosophila* testis cells caused a decrease in male fecundity, a decrease in the number of testis Tj-positive cells, and a change in the level of AHR target gene transcriptions. We concluded that exposure to AHR ligands could potentially lead to the risk of male infertility. Notably, we also found that the activation in the expression of some AHR target genes depends on the expression of testis-specific chaperone CG5017 in gonadal cells. Since CG5017 belongs to the nucleosome assembly protein (NAP) family [34] and may participate in epigenetic regulation, we proposed that this nucleotropic chaperone is essential to provide human AHR access to a defined set (but not all) of its target genes in soma during spermatogenesis.

## 2. Results and Discussion

### 2.1. The Effect of AHR Exogenous Ligands on Male Fecundity

We have previously demonstrated that *Drosophila* may serve as a valid model organism to investigate the complex effects of xenobiotics on human AHR functioning in vivo [32]. In order to investigate the effects of xenobiotics on male fecundity, we used *Drosophila* males carrying inducible human *AhR* (the *UAS-AhR* construct is described in Reference [32]) and *Tj-GAL4* drivers. In *Drosophila* testes, the *Tj-GAL4* driver activates *UAS*-constructs in cells which are in connection with generative cells. We refused to use the *Nos-GAL4* driver, which activates the *UAS*-constructs in germ-line cells since the fertility of *UAS-AhR/Nos-GAL4* males was low even without exposure to exogenous ligands (about 50% according to our unpublished data, *n* = 46). This indicates the presence of endogenous AHR ligands capable of activating human AHR in *Drosophila Nos-*positive cells that could potentially falsify experimental results. The fertility of *UAS-AhR/Tj-GAL4* males raised on standard nutrient medium or fed with xenobiotic was not disturbed (100%, *n* = 50). It allowed us to study the effect of xenobiotic-mediated AHR activation on male fecundity using *Tj-Gal4* driver.

The fecundity of *UAS-AhR/Tj-GAL4* males fed with xenobiotics and of control *UAS-AhR/Tj-GAL4* males (without exposure to xenobiotic) was measured by mating them to wild-type *Oregon R* females and counting the number of undeveloped eggs produced per female over a four-day period. The replacement of fertilized females with virgins was performed daily. The effects of exogenous ligands on *UAS-AhR/Tj-GAL4* males resulted in an increase in the proportion of undeveloped eggs in the first two days after ligand action (Figure 1). Most of the effect is caused by indirubin and beta-Naphthoflavone in the first two days after ligand exposure. Remarkably, these effects are completely reversible; when flies fed with the xenobiotic are shifted back to a standard diet, male fecundity is rapidly restored. The mechanism by which it is restored is not clear yet.

Most likely, the decline in male fecundity was caused by the disturbances in testes cells involved in the formation of the functional spermatozoa. In the *Drosophila* testis, germ-line stem cells and progenitor somatic stem cells reside at the tip of the testis, known as the apical hub [35]. Tj positive cells are important for the differentiation of the germ-line [35]. As we used a *Tj-Gal4* driver to generate AHR misexpression, we proposed that the reason for the decrease in fecundity of *Tj-GAL4/UAS-AhR* males in response to exogenous ligands might be due to the disruptions in division of Tj-positive cells.

To test our hypothesis, we estimated the number of Tj-positive cells in testes of *UAS-AhR/Tj-GAL4* flies fed with AHR exogenous ligands for two days, also using control *UAS-AhR/Tj-GAL4* flies developed on a standard medium. We found that the testes of flies fed with xenobiotics were thinner (Figure 2), and a decrease in the average number of Tj-positive cells per testis was observed in flies fed with xenobiotics for 3 days (indirubin 75.8 ± 6.19; *n* = 18, beta-Naphthoflavone: 86 ± 9.7; *n* = 13, indinol: 79.8 ± 8.1; *n* = 14) when compared to testes from males raised on the standard medium (106.3 ± 8.25; *n* = 24) (Figure 2E). No remarkable differences between testis of *Tj-GAL4/+* flies fed with xenobiotic and testis of flies with the same genotype developed on standard medium were detected (Appendix A
Figure A1). Thinner testes were typical for only *UAS-AhR/Tj-GAL4* flies fed with xenobiotic so we attributed this effect to the ectopic AHR activation.

In *Drosophila* testis, the absence of Tj-positive cells blocks normal spermatogenesis [36]. Thus, the decrease in the number of Tj-positive cells in response to human AHR activation by exogenous ligands in testes of *UAS-AhR/Tj-GAL4* flies could be the reason of a reduced production of spermatozoa and decreased male fecundity.

We believe that the cause of the detected functional and morphological differences between the control and experimental males should be due to the activities of the AHR targeted genes that regulate homeostasis and cell division. 

### 2.2. The Effects of Exogenous Ligands and Testis-Specific Chaperone CG5017 on the Expression of AHR Target Genes in Drosophila Testes

To assess the ability of xenobiotics to influence the expression of human AHR target genes in *Drosophila* testes, we first identified potential human AHR target genes in *Drosophila* (described in Reference [32]). We selected several putative *Drosophila* homologs of human AHR targets genes containing *XRE*-elements in their regulatory regions: *Mannosyl (α-1,3-)-glycoprotein β-1,2-N-acetylglucosaminyltransferase 1* (*Mgat1*), which participates in the determination of adult lifespan relating to mushroom body development; *Glutathione S transferase T4* (*GstT4*), which is involved in oxidation-reduction processes and catalyzes reactions of biotransformation; *Cytochrome P450 6g1* (*Cyp6g1*), which is involved in the oxidation-reduction process, response to DDT, and the insecticide catabolic process; *N-acetylneuraminic acid synthase* (*Nans*), which participates in the carbohydrate biosynthetic process; *Relish* (*Rel*), which encodes the NF-κB subunit; *p53*, which is a transcriptional factor required for adaptive responses to genotoxic stresses, including cell death, compensatory proliferation and DNA repair; *Myc,* a transcription factor related to proto-oncogenes, which contributes to cell growth, cell competition, and regenerative proliferation; *dаcapo* (*dap*), which encodes the Cyclin-dependent kinase inhibitor; the *Retinoblastoma-family protein* (*Rbf*), which provides negative regulation of the G1/S transition of mitotic cell cycles; *Jun-related antigen* (*Jra*), which is involved in positive regulation of the metabolic process, humoral immune response, aging, and RNA polymerase II transcription factor activity; and *Dcdc42* (*Cdc42*), which is a key regulator of the actin cytoskeleton, playing a central role in actin cytoskeleton organization, morphogenesis, hemocyte migration, cell polarity, and wound repair.

To investigate the effects of exogenous ligands in vivo, we analyzed the expression of AHR target genes by RT-PCR in testes of *UAS-AhR/Tj-GAL4* flies fed with xenobiotics for two days. To activate human AHR, we used exogenous ligands known to act as agonists of this receptor. This means that these molecules only cause an increase in the transcription levels of AHR target genes [20,37]. We found that induced human AHR had pleiotropic effects on its target genes, depending on the nature of the exogenous ligand. In other words, the xenobiotic-mediated effect of human AHR activity in testes of *UAS-AhR/Tj-GAL4* flies resulted in three different ways: Some experienced a decrease in the gene expression, some an increase in gene expression, and several genes had no response to AHR activity (Figure 3). For example, the activation of human AHR by indirubin resulted in the activation of almost all genes tested except *Mgat1, Cyp6g1*, and *Myc*. The activation of human AHR by beta-Naphthoflavone resulted in the activation of *Cyp6g, Rel,* and *Myc*, and the suppression of *Mgat1* and *dap* genes. The activation of the human AHR by indinol resulted in the suppression of *Cyp6g* and the weak activation of *Mgat1, GstT4, Csas, Rel, p53, Myc,* and *Jra* genes. 

In our previous study, a similar effect was found [32]. We attributed this effect to the epigenetic repressive state of chromatin, which limits the ability of a human AHR to access *XREs* and control its target gene expression in the *Drosophila* genome. This hypothesis was confirmed in our experiments, through which we demonstrated that the effects of exogenous ligands on AHR target genes are mediated by the polycomb group (PcG) epigenetic chromatin regulators [32]. 

The formation of the epigenetic state of genes involved not only Pc and Trx complexes, but also NAP family nucleotropic chaperones which control the activity of H2A-H2B histones [34,37,38,39,40]. It was previously shown that NAP family *CG5017* and *spineless* (ss, *D. melanogaster* homologue of mammalian *AhR*) act synergistically, controlling morphogenesis, memory, and detoxification [41,42]. The synergy in the genetic interactions between hypomorphic mutations of *ss* and CG*5017* may reflect the involvement of NAP family chaperones in the regulation of AHR-signaling in *D. melanogaster.* We decided to study the effect of CG5017 on human AHR target gene transcription in somatic cells of *Drosophila* testis. To test this, we performed experiments using *UAS-AhR/Tj-GAL4* flies carrying a mutant hypomorphic allele of *CG5017*-*ss^aSc^* [43,44]. To activate human AHR in *UAS-AhR/Tj-GAL4; ss^aSc^* flies we added indirubin, beta-Naphthoflavone, and indinol into the nutrient medium. The mRNA levels of AHR target genes were measured by RT-PCR in testes of *UAS-AhR/Tj-GAL4; ss^aSc^* flies fed with exogenous ligands for 2 days. Flies of the same genotype developed on the standard nutrient medium were used as a control. A remarkable increase in the transcription of some AHR target genes was observed (Figure 4). 

Activation of AHR by xenobiotics on the background of a mutant allele of *CG5017* omitted the silencing of AHR target genes involved in maintaining cell homeostasis (Table 1). For example, the activation of human AHR by indirubin resulted in strong activation of *Cyp6g1* (up to 22 times). The activation of human AHR by beta-Naphthoflavone de-repressed *Mgat1* (up to 3–5 times), and omitted the silencing of *GstT4, Csas* and *Nans.* The activation of human AHR by indinol is not very pronounced and resulted in de-repression of *Cyp6g1.* On the other hand, the transcription levels of genes regulating cell proliferation and differentiation were either not affected or were decreased (Figure 4, Table 1).

The regulatory mechanism of AHR target gene expression by *CG5017* is not clear. Since *CG5017* belongs to the nucleosome assembly protein (NAP) family [34] and may participate in epigenetic regulation, we proposed that this nucleotropic chaperone could be essential to enable human AHR to access only a defined set of its target genes in soma during spermatogenesis. 

Our results indicate a complex, multi-step regulatory mechanism of proper AHR target gene transcription which can be disrupted by exogenous ligands, which may be the cause of many diseases [45]. We hope that further study of the exogenous ligands’ action mechanisms will help in the development of strategies for limiting xenobiotic effects and reducing pathology.

## 3. Materials and Methods 

### 3.1. Fly Stocks, Rearing Conditions, Reagents and Crosses

*UAS-AhR* strain with inducible human *AhR* gene expression in *D. melanogaster* genome was generated early [32]. Wild type *Oregon R* and *Tj-GAL4/Cy* strains were obtained from Bloomington *Drosophila* stock center. Also, we used *ss^aSc^* strain with hypomorphic mutations of *CG5017* and *spineless* genes [43,44].

Flies were reared on nutrient Formula 4-24 Instant Drosophila Medium (Carolina Biological Supply, Burlington, NC, USA). Following ligands were used: 2′Z-Indirubin (Sigma-Aldrich, St. Louis, MO, USA), beta-Naphthoflavone (Thermo Fisher Scientific, Waltham, MA, USA), Indole-3-Carbinol (Mirax Biopharma, Moscow, Russia). Ligand solutions were prepared as described in [32]. Final concentrations of beta-Naphthoflavone, indirubin and indole-3-carbinol were 200 µg/g medium, 25 µg/g medium, 10 µg/g medium correspondently. 

Ligands were fed to imago F1 offspring after the crossing of *Tj*-*GAL4* males with *UAS-AhR* females. Parents were kept on standard Formula 4-24 medium. After hatching flies of first day old were selected for feeding experiments. Flies were kept at room temperature (25 °C). 

To obtain flies of *UAS-AhR/Tj-GAL4; ss^aSc^* genotype we crossed *UAS-AhR/Cy; ss^aSc^/D* females with *Tj-GAL4/Cy*; *ss^aSc^/D* males and flies without balancer chromosomes were further selected in the F1 offspring. Flies were kept at room temperature (22 °C).

### 3.2. Calculation of Undeveloped Eggs Frequency

Reproductive output was measured at 25 °C. Imago males *Tj-GAL4/UAS-AhR* (*n* = 7) were fed with ligands solutions for 2 days whereupon males were crossed with *Oregon R* virgin females (*n* = 7) in fresh medium vials. During four days we replaced fertilized *Oregon R* females with virgin ones after 24 hr and counted the total number of eggs laid. We considered unfertilized eggs that did not develop within 24-25 hr. The experiment was performed three times. The proportion of daily undeveloped eggs per female was calculated using the following formula: [(Number of undeveloped eggs/Total number of eggs) × 100%]/[Number of females tested]. 

### 3.3. Real-Time Reverse-Transcription PCR Analysis

Experiments have been done in triplicate as described previously [32]. Primers and TaqMan^®^ probes used for RT-qPCR experiments are available in Appendix A
Table A1.

### 3.4. Immunohistochemistry 

Experiments have been done as described previously [32]. Primary antibody used in our work is guinea-pig polyclonal anti-Tj (1:5000) [46]. Secondary antibodies (1:200) were conjugated to Alexa Fluor–488 (Molecular Probes, Waltham, MA, USA). DNA was stained with SytoxGreen (1:500, Thermo Fisher Scientific, Waltham, MA, USA).

### 3.5. Microscopic Analysis

The resulting immunostaining preparations were examined using Leica TCS SP5 confocal microscope using a multichannel mode with a 40 × immersion oil lens. The images were recorded with a z-resolution of 0.7–0.8 μm.

### 3.6. Image Analysis

The resulting images were imported into Imaris^®^ 5.0.1 (Bitplane AG, Belfast, UK) for further processing. Estimation of somatic cells on the confocal images was carried out by measuring number of Tj-stained cells. Student’s *t*-tests were used confirmation of statistical significance. The threshold of statistical significance was *p* ≤ 0.01 for beta-Naphthoflavone and *p* ≤ 0.001 for indirubin and indinol.

## Figures and Tables

**Figure 1 ijms-20-00118-f001:**
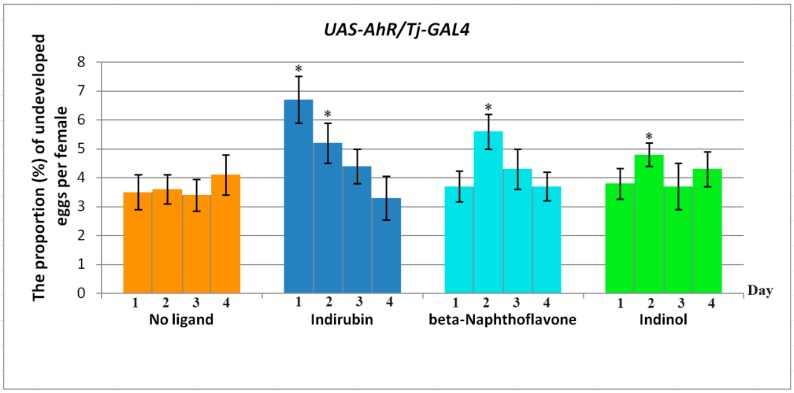
Daily effect of exogenous ligands on the proportion of undeveloped eggs from wild-type female after crossing with *UAS-AhR/Tj-GAL4* male without exposure to ligands (control, orange), exposed to indirubin (blue), beta-Naphthoflavone (azure), indinol (green). Data correspond to the mean ± SD of three independent experiments. Asterisks mean the significant difference compared to the control group. Statistical analysis was performed using Student’s *t*-test (* *p* ≤ 0.05).

**Figure 2 ijms-20-00118-f002:**
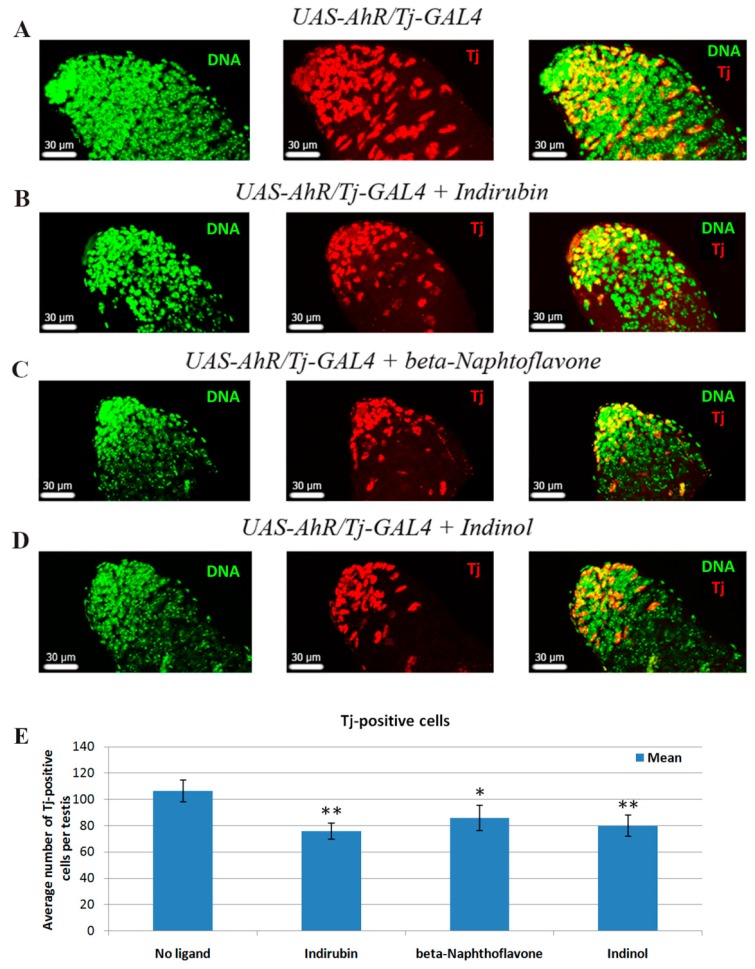
Xenobiotic-mediated activation of human AHR in cells of *Drosophila* testis causes a decrease in number of Tj-positive cells. (**A**–**D**) Confocal immunofluorescence sections of the apical tip of testes stained with SytoxGreen to highlight DNA (green) and anti-Tj to visualize Tj-positive cells (red). The third column represents merged images. (**A**) Testes are from flies raised on standard medium or fed for 3 days with (**B**) indirubin, (**C**) beta-Naphthoflavone, (**D**) indinol. All tested males were of the same *UAS-AhR/Tj-Gal4* genotype. Note that apical tips of males fed with xenobiotics are smaller than apical tip of male fed with standard medium. Scale bars, 30 µm. (**E**) Quantification of Tj-positive cells in testes of *UAS-AhR/Tj-GAL4* flies raised on standard medium (No ligand) and on medium with indirubin, beta-Naphthoflavone and indinol for 3 days counted at the 4^th^ day after feeding. Error bars represent 90% confidence interval of the mean. Asterisks indicate statistically significant difference using Student’s *t*-test (* *p* ≤ 0.01; ** *p* ≤ 0.001).

**Figure 3 ijms-20-00118-f003:**
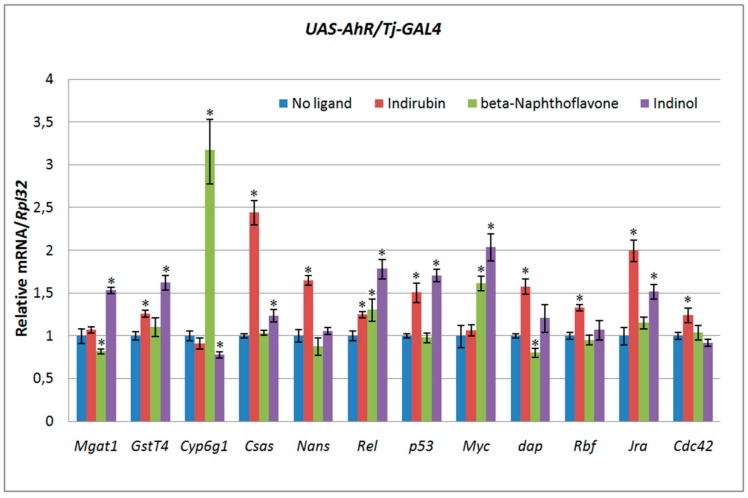
Different effects of human AHR exogenous ligands on AHR target gene mRNA levels in testes of *UAS-AhR/Tj-GAL4* flies. Relative mRNA levels were analyzed by real-time PCR in testes dissected from *UAS-AhR/Tj-GAL4* flies fed with indirubin, beta-Naphthoflavone or indinol for 2 days. *UAS-AhR/Tj-GAL4* flies developed on standard medium were used as a control. Transcript levels are represented as means ± SD (error bars). * *p* < 0.05, compared to the control. Statistical analysis was performed using Student’s *t*-test.

**Figure 4 ijms-20-00118-f004:**
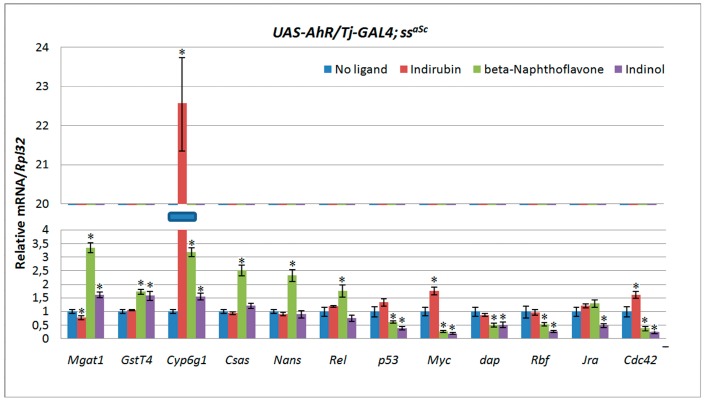
Decreased expression of CG5017 nucleotropic chaperone leads to ligand dependent activation in transcription of some AHR target genes. Relative mRNA levels were analyzed by real-time PCR in testes dissected from *UAS-AhR/Tj-GAL4*; *ss^aSc^* flies fed with indirubin, beta-Naphthoflavone or indinol for 2 days. *UAS-AhR/Tj-GAL4; ss^aSc^* flies developed on standard medium were used as a control. Transcript levels are represented as means ± SD (error bars). * *p* < 0.05, compared to the control. Statistical analysis was performed using Student’s *t*-test.

**Table 1 ijms-20-00118-t001:** The decreased expression of nucleotropic chaperone CG5017 activates ligand-dependent transcription of some AHR target genes. Summarized results of real-time PCR experiments shown on Figure 3 and Figure 4. «*+/+»* and *«ss^aSc^*» columns represent results shown on Figure 3 and Figure 4 respectively. «+», «−» and «0» mean the increasing expression, the decreasing expression and no effect, respectively. Red pluses mean the remarkable increasing in transcription on the background of mutant *CG5017.*

Gene Symbol	Ligand					
	Indirubin		beta-Naphthoflavone		Indinol	
	Allele of *CG5017*					
	+/+	*ss^aSc^*	+/+	*ss^aSc^*	+/+	*ss^aSc^*
***Mgat1***	0	−	−	+	+	+
***GstT4***	+	0	0	+	+	+
***Cyp6g1***	0	+	+	+	−	+
***Csas***	+	0	0	+	+	0
***Nans***	+	0	0	+	0	0
***Rel***	+	0	+	+	+	0
***p53***	+	0	0	−	+	−
***Myc***	0	+	+	−	+	−
***dap***	+	0	−	−	0	−
***Rbf***	+	0	0	−	0	−
***Jra***	+	0	0	0	+	−
***Cdc42***	+	+	0	−	0	−

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
