# Peer review of "NAP Family CG5017 Chaperone Pleiotropically Regulates Human AHR Target Genes Expression in Drosophila Testis"

_ijms, 2018, doi:10.3390/ijms20010118_

Reviewer 1 Report

In this manuscript, the authors investigate the effects of human AHR ligands on male fecundity using Drosophila expressing ectopically the human Ahr gene in testicular somatic cells. They reported the generation of “humanized” Drosophila transgenic flies carrying the inducible human AhR gene in a previous study and found that the effect of xenobiotics on AHR target expression was more pronounced in organs with a high number of proliferating cells. They therefore hypothesized that ectopic AHR expression induced by xenobiotics in testes may be the cause of various disturbance of spermatogenesis. In the present study, they found that exogenous AHR ligands (xenobiotics) decrease male fecundity, decrease the number of gonadal Tj-positive cells and lead to both increased or decreased AHR target genes expression. They also found that the expression of some AHR target genes depends on the expression of CG5017, a testis-specific chaperone that may participate in epigenetic regulation. The authors suggest that gonadal AHR activation may affect the expression of gene networks that control sperm production and that CG5017 is essential to provide to the human AHR the access to a defined set of target genes during spermatogenesis. The manuscript is clear, well written, and the data presented in this study are novel and interesting. However, modifications of the manuscript are required:

1. Please remove “epigenetic regulation” in the title of the manuscript since there is no evidence that AHR target gene expression is epigenetically regulated by CG5017 in the present study.

2. Lines 20, 79, 191, 245: Please replace “same” by “some”

3. Table 1 and Figure 1 represent the same results. Please choose to represent the data in Table 1 or in Figure 1.

4. There are no error bars in the data presented in Figure 1. Has this experiment been repeated several times? If not, please show the data from at least three independent experiments.

5. Lines 98-99: “The effects of exogenous ligands on UAS-AhR/Tj-GAL4 males resulted in an increase in the proportion of undeveloped eggs”. Is there a significant increase in the proportion of undeveloped eggs? Significant differences are not represented in Figure 1. 

6. Line 120: please replace “visualized” by “visualize”

7. In addition to the decrease in the number of somatic Tj-positive cells, is there a significant decrease in the number of germ cells and in the production of spermatozoa in UAS-AhR/Tj-GAL4 males exposed to xenobiotics?

8. Lines 161-165: “the activation of human AHR by indirubin resulted in the activation of almost all genes tested except Mgat1, Cyp6g1 and Myc. The activation of human AHR by beta-Naphthoflavone resulted in the activation of Cyp6g, Rel, Myc and the suppression of Mgat1 and dap genes. The activation of the human AHR by indinol resulted in the suppression of Cyp6g and the weak activation of Mgat1, GstT4, Csas, Rel, p53, Myc, and Jra genes”. Are the activation or suppression of gene expression statistically significant? Significant differences are not represented in Figure 4.

9. Line 167: please replace “transcription” by “mRNA levels”

10. Line 186: Please replace “hypomorpfic” by “hypomorphic”

11. Line 191: “remarkable increasing in transcription”. Is the increase in mRNA levels significantly significant? Significant differences are not represented in Figure 5.

12. Lines 190-191 and lines 199-202: “The remarkable increasing in transcription of same AHR target genes was seen (Figure 5). The increase in transcription levels of different genes in response to different ligands is different. The increased level of transcription is mainly noted for genes involved in the maintaining of cell homeostasis. The transcription level of genes regulating cell proliferation and differentiation was either not affected or decreased”. These four sentences are not clear. Please explain. 

13. Lines 222-226: “To activate human AHR by exogenous ligands, we used only molecules which are known to act as agonists of this receptor, i.e. cause only an increase in the transcription levels of AHR target genes in mammals [20,35]. In addition to an increase, we found unchanged expression levels of the Drosophila AHR target genes. It was not unexpected result because in our previous study a similar effect was found [32].” The same sentences are found earlier in the manuscript (lines 172-176). Please remove these repetitions. 

14. Lines 236 and 240: please replace “hylomorphic” by “hypomorphic”

15. Lines 264-265: please replace “mkg/g” by “mg/g”

16. Please explain how the concentrations of xenobiotics were chosen.

Author Response

We thank all reviewers for their comments and their advice on how to improve the manuscript. We have now modified the manuscript in accordance to the reviewers’ suggestions and answered all of their concerns. Detailed answers to specific comments are below.

Reviewer 2 Report

In this study, Akishina et al. investigate the effect of human AHR ligands on Drosophila male fecundity. The authors show that exogenous ligands may affect the expression of AHR target genes. Besides, they show that the ligand-dependent activation of some AHR target genes depends on the expression of testis-specific NAP family chaperone CG5017.

The study is interesting in the perspective of understanding the function of AHR in vivo rather than in cultured cells and uses a “humanised” transgenic fly strain that allows for the expression of human AHR in Drosophila testis. This allows for the estimation of possible effects of exogenous AHR ligands on spermatogenesis.

Although the study is interesting, I found several weak points which, if reinforced, would strengthen the conclusions of the paper.

Major points:

- Figure 1, Table 1: it appears that mostly indirubin - and beta-Naphthoflavone to a lesser extent - affects the proportion of undeveloped egg; however, the experiment has only been repeated twice, and no statistical significance is shown. The authors should indicate the outcome of the experiment as mean ± SD and give the statistical significance of the results.

- Figure 2: the apical tip of males fed with xenobiotics appear to be smaller than the ones from flies fed on a non-supplemented diet. Can this effect be quantified? Is the reduction of the apical tip dependent on the expression of exogenous AHR? What happens when flies Tj-GAL4/+ are fed with the xenobiotics vs non-supplemented diet?

- What happens to the proportion of Tj cells in UAS-AHR/Tj-GAL4;ssaSc flies ± xenobiotics?

- L.214: to support the conclusion that “a decrease in the number of Tj‐positive cells of testes is likely to be the main reason of defective spermatozoa that cannot provide fertilisation” the authors could try to generate males that lack Tj cells using a UAS-hid transgene and evaluate the fecundity of the flies.

- Is CG5017 expressed exclusively in the testis? If not, the author should check that the effects on gene expression observed in the ssaSc mutant are specific of the lack of function in testis. If available, silencing of CG5017 with RNAi in Tj cells should be performed concomitantly to AHR expression ± xenobiotics.

- What is the impact of CG5017 lack-of-function on the fecundity of the males?

Minor points:

- OregonR females were mated with UAS-AhR/Tj-GAL4 males. Are the proportions of undeveloped eggs affected by the genetic background of the flies? What would happen if using Canton-S or other wild-type strains?

- Overexpression of AHR alone has severe developmental and morphogenesis effects on other tissues; is there any defect induced by the expression of UAS-AHR using Tj-GAL4?

- Is it possible to evaluate the effect of the xenobiotics on the expression of human AHR (RT-qPCR? Western blotting?)? Effect on endogenous Drosophila AHR/spineless?

- Figures 3 and 4: is the proportion of Tj cells restored over time when the males are not being fed with the xenobiotics anymore? Figure 1 shows that the proportion of undeveloped eggs return to a standard rate after 4 days, does that correlate with a restoration of the proportion of Tj cells?

- How is the expression of AHR target genes in ssaSc mutant flies compared to wild-type flies?

- L.127-128: the sentence “[…] for 3 days counted at 1 day after feeding” is confusing.

Author Response

We thank all reviewers for their comments and their advice on how to improve the manuscript. We have now modified the manuscript in accordance to the reviewers’ suggestions and answered all of their concerns. Detailed answers to specific comments are below.

Reviewer #2

- “Figure 1, Table 1: it appears that mostly indirubin - and beta-Naphthoflavone to a lesser extent - affects the proportion of undeveloped egg; however, the experiment has only been repeated twice, and no statistical significance is shown. The authors should indicate the outcome of the experiment as mean ± SD and give the statistical significance of the results.

Answer. We added error bars of data received from three independent experiments, performed statistic analysis and marked the significant changes by asterisk.

- “Figure 2: the apical tip of males fed with xenobiotics appear to be smaller than the ones from flies fed on a non-supplemented diet. Can this effect be quantified? Is the reduction of the apical tip dependent on the expression of exogenous AHR? What happens when flies Tj-GAL4/+ are fed with the xenobiotics vs non-supplemented diet?

Answer. The sizes of apical tips of Tj-GAL4/+ flies fed with xenobiotic or developed on standard medium were the same and any decrease in a number of Tj-positive cells was not observed (104.8±7.7; n=19 and 100.1±5.2; n=22 correspondently)We noticed that thinner testes were typical for only UAS-AhR/Tj-GAL4 flies fed with xenobiotic so we attributed this effect to the ectopic AHR activation. 

- “What happens to the proportion of Tj cells in UAS-AHR/Tj-GAL4; ssaSc flies ± xenobiotics?

Answer. In UAS-AHR/Tj-GAL4 flies on the background of hypomorphic allele of ssaSc we observed the decrease in a number of Tj-positive cells after their stimulation by xenobiotic. This effect was analogous of that observed in UAS-AHR/Tj-GAL4 flies. But in UAS-AHR/Tj-GAL4 flies, AHR target gene expression were not remarkable increased in response to xenobiotic exposure so we decided to test what happens with AHR target gene expression in flies with mutant allele of CG5017 chaperon.

 - “L.214: to support the conclusion that “a decrease in the number of Tj‐positive cells of testes is likely to be the main reason of defective spermatozoa that cannot provide fertilisation” the authors could try to generate males that lack Tj cells using a UAS-hid transgene and evaluate the fecundity of the flies.

Answer. According to: Li, et al., 2003. [The large Maf factor Traffic Jam controls gonad morphogenesis in Drosophila. Nature Cell Biology, 5(11), 994–1000] the absence of Tj-positive cells disrupted gonad morphogenesis.

- “Is CG5017 expressed exclusively in the testis? If not, the author should check that the effects on gene expression observed in the ssaSc mutant are specific of the lack of function in testis. If available, silencing of CG5017 with RNAi in Tj cells should be performed concomitantly to AHR expression ± xenobiotics.”

Answer. CG5017 encodes testis-specific chaperon as described in:  Kimura, S. (2013). The Nap family proteins, CG5017/Hanabi and Nap1, are essential for Drosophila spermiogenesis. FEBS Letters, 587(7), 922–929.

- “What is the impact of CG5017 lack-of-function on the fecundity of the males?

Answer. Cited above Kimura wrote that “Adult homozygous hanabi mutant males were completely sterile”. That is why we used hypomorphic allele of this gene.

 - “Oregon R females were mated with UAS-AhR/Tj-GAL4 males. Are the proportions of undeveloped eggs affected by the genetic background of the flies? What would happen if using Canton-S or other wild-type strains?

Answer. Wild-type (Oregon R or Canton S) females usually put some amount of undeveloped eggs. To avoid this background, we made control crossing of wild-type females with UAS-AhR/Tj-GAL4 males developed on standard medium and compare it to the crossing of wild-type females with males of the same genotype fed with xenobiotic, so the final proportion of undeveloped eggs was not affected.

-” Overexpression of AHR alone has severe developmental and morphogenesis effects on other tissues; is there any defect induced by the expression of UAS-AHR using Tj-GAL4?

Answer. We found the sterility using Nos-GAL4 driver to induce overexpression of AHR in male germ-line cells, and so we refused to use it in our experiments. Any defects induced by Tj-Gal4 in UAS-AhR/Tj-GAL4 males were not observed.

- “Is it possible to evaluate the effect of the xenobiotics on the expression of human AHR (RT-qPCR? Western blotting?)? Effect on endogenous Drosophila AHR/spineless?

Answer. We provided Western blot analysis of inducible human AHR expression and RT-qPCR of its mRNA in D. melanogaster in our previous article: Akishina et al., 2017; supplemental material SpFig. 5.

 It is known that xenobiotics do not activate invertebrate homologous of AHR (Duncan et al., 2002; Céspedes et al., 2010).

- “Figures 3 and 4: is the proportion of Tj cells restored over time when the males are not being fed with the xenobiotics anymore? Figure 1 shows that the proportion of undeveloped eggs return to a standard rate after 4 days, does that correlate with a restoration of the proportion of Tj cells?”

Answer. We did not perform such experiment, but we believe that the proportion of Tj-positive cells will restore. 

- “How is the expression of AHR target genes in ssaSc mutant flies compared to wild-type flies?” 

Answer. According to our data AHR target genes expression in ssaSc mutants is lower than in wild-type flies (Kuzin et al., 2014). That is why we compared the levels of AHR target genes in flies with the same UAS-AhR/Gal4-Tj genotype (before and after their stimulation by xenobiotic) in the ssaSc mutant background (fig. 3, 4).

- “L.127-128: the sentence “[…] for 3 days counted at 1 day after feeding” is confusing”.

Answer. The sentence is edited.

Reviewer 3 Report

In this study, the authors investigated the effects of human AHR ligands on male fecundity in Drosophila melanogaster. This study utilized UAS-GAL4 system to induce human AhR expression in Drosophila and found that exogenous AHR ligands decrease the number of Drosophila gonadal Tj-positive cells. Using RT-PCR, the authors examined the expression of 12 AHR target genes under AHR ligands treatment; and found that some of the target genes are up-regulated while the others are down-regulated. In addition, the authors also found that the above regulations are affected by CG5017 reduction. In the current manuscript, significance tests are missing, the RT-PCR data are not fully discussed, the epigenetic regulation part are overinterpreted. In all, the manuscript is premature that prevent its publication in the current version.

Major concerns:

1. In Fig. 1, significance tests are missing. It’s barely to tell the difference regarding to 3.5% in control vs. 6.7% in Indirubin induction.

2. Fig. 3 is the quantification for Fig. 2, thus Fig.2 and 3 should be combined in one Figure.

3. In Fig.4 and 5, significance tests are missing. 

4. “CG5017 participate in epigenetic regulationof AHR target genes expression” is only a hypothesis, the authors did not show any epigenetic evidence for this proposal, involving epigenetic marker or chromatin structure. Thus, the title and abstract should be revised according to the existing data. 

5. What’s the biological implication of the different changes of AHR target genes expression before and after CG5017 reduction? Considering some genes are affected while the other genes are not affected upon CG5017 reduction. The authors only showed the changes without fully discussed them.

6. Discussion and Results sections are almost the same. 

Author Response

Answers to the reviewers’ comments

We thank all reviewers for their comments and their advice on how to improve the manuscript. We have now modified the manuscript in accordance to the reviewers’ suggestions and answered all of their concerns. Detailed answers to specific comments are below.

Reviewer #3

1. “In Fig. 1, significance tests are missing. It’s barely to tell the difference regarding to 3.5% in control vs. 6.7% in Indirubin induction.

Answer. We improved this.

2. “Fig. 3 is the quantification for Fig. 2, thus Fig.2 and 3 should be combined in one Figure.

Answer. We improved this.

3. “In Fig.4 and 5, significance tests are missing”. 

Answer. We improved this.

4. “CG5017 participate in epigenetic regulation of AHR target genes expression” is only a hypothesis, the authors did not show any epigenetic evidence for this proposal, involving epigenetic marker or chromatin structure. Thus, the title and abstract should be revised according to the existing data.” 

Answer. We improved this: Effect of NAP Family CG5017 Hypomorphic Mutation on Human AHR Target Genes Expression in Drosophila Testis.

5. “What’s the biological implication of the different changes of AHR target genes expression before and after CG5017 reduction? Considering some genes are affected while the other genes are not affected upon CG5017 reduction. The authors only showed the changes without fully discussed them.

Answer. The data we provided in our manuscript are preliminary, so today we are not ready to discuss it in details. The biological implication of the demonstrated phenomena we believe to get in soon coming results. That is why we edited manuscript from article to short communication.

6. “Discussion and Results sections are almost the same.” 

Answer. We improved this.

Round  2

Reviewer 1 Report

The authors addressed all my recommendations and comments. The manuscript is suitable for publication after minor revisions:

1. Lines 140-142: “the decrease in the number of Tj-positive cells in response of human AHR activation by exogenous ligands in testes of UAS-AhR/Tj-GAL4 flies could be the reason of non-functional spermatozoa…”

Please remove “non-functional spermatozoa”. Indeed, the authors do not provide evidence that spermatozoa are non-functional. Another hypothesis for the decrease in male fecundity could be a reduced production of spermatozoa.

2. The English writing of the manuscript still needs to be improved. Please proofread the entire manuscript carefully.

Author Response

We are grateful to the reviewers for the positive assessment of our manuscript revision. Minor revisions in accordance to the reviewers’ suggestions were made. English language was edited by MDPI English Editing Service.  Answers to specific comments are below.

1. “Lines 140-142: “the decrease in the number of Tj-positive cells in response of human AHR activation by exogenous ligands in testes of UAS-AhR/Tj-GAL4 flies could be the reason of non-functional spermatozoa…”

Please remove “non-functional spermatozoa”. Indeed, the authors do not provide evidence that spermatozoa are non-functional. Another hypothesis for the decrease in male fecundity could be a reduced production of spermatozoa”

Answer. The sentence was edited.

2. “The English writing of the manuscript still needs to be improved. Please proofread the entire manuscript carefully.”

Answer. We inform you that the manuscript was edited by MDPI English Editing Service.

Reviewer 2 Report

The authors have made a significant work on editing the manuscript and have notably chosen to merged the Results and Discussion section.

The authors have provided answers to all my previous questions.

Hereafter are some comments on the new version of the manuscript.

- Figure 2: from their answer to my question regarding the size of the apical tips, it appears the authors have done some experiments in Tj-GAL4/+ as a negative control. Would it be possible to add it as supplementary data?

- Figures 2: error bar should show the standard error rather than a 90% confidence interval to be consistent with error representation in Figure 1.

- Figures 3 and 4: what statistics test was used to proceed to the analysis? What is the P-value used as a threshold in these case?

- The validation of the effect of the xenobiotics on UAS-AHR expression previously published used a head-specific driver (Elav-GAL4). It would be more accurate to show the expression with the Tj-GAL4 driver used in the present study, if available. If not possible, maybe the authors could mention in the manuscript that the previous report hasn't shown any effect on ectopically expressed AHR.

- line 110: font size to adjust

- line 210: font size to adjust

Author Response

We are grateful to the reviewers for the positive assessment of our manuscript revision. Minor revisions in accordance to the reviewers’ suggestions were made. English language was edited by MDPI English Editing Service.  Answers to specific comments are below.

- “Figure 2: from their answer to my question regarding the size of the apical tips, it appears the authors have done some experiments in Tj-GAL4/+ as a negative control. Would it be possible to add it as supplementary data?”

Answer. The supplementary data was added. We also added following sentences to the main text: “No remarkable differences between testis of Tj-GAL4/+ flies fed with xenobiotic and testis of flies with the same genotype developed on standard medium were detected (Supplementary Figure A1). Thinner testes were typical for only UAS-AhR/Tj-GAL4 flies fed with xenobiotic so we attributed this effect to the ectopic AHR activation.”

- “Figures 2: error bar should show the standard error rather than a 90% confidence interval to be consistent with error representation in Figure 1.”

Answer. We used Confidence intervals (Figure 2) because we had a large sample (more than 10) and according to: [Cumming, G., Fidler, F., & Vaux, DL (2007). Error bars in experimental biology. The Journal of Cell Biology, 177 (1), 7–11.] it is better to use Confidence intervals in this case. Whereas, the average for three independent replications is displayed on Figure 1, and in this case, it is indeed more expedient to use SD.

- “Figures 3 and 4: what statistics test was used to proceed to the analysis? What is the P-value used as a threshold in these cases?”

Answer. To indicate the statistically significant difference between means we used Student’s t-test in all our experiments. P-value was added to Figure legends.

- “The validation of the effect of the xenobiotics on UAS-AHR expression previously published used a head-specific driver (Elav-GAL4). It would be more accurate to show the expression with the Tj-GAL4 driver used in the present study, if available. If not possible, maybe the authors could mention in the manuscript that the previous report hasn't shown any effect on ectopically expressed AHR.”

Answer. We mentioned in our manuscript (line 92-96) that: “We refused to use the Nos-GAL4 driver, which activates the UAS-constructs in germ-line cells since the fertility of UAS-AhR/Nos-GAL4 males was low even without exposure to exogenous ligands (about 50% according to our unpublished data, n=46). This indicates the presence of endogenous AHR ligands capable of activating human AHR in Drosophila Nos-positive cells that could potentially falsify experimental results.”

“The fertility of UAS-AhR/Tj-GAL4 males raised on standard nutrient medium or fed with xenobiotic was not disturbed (100%, n=50). It allowed us to study the effect of xenobiotic-mediated AHR activation on male fecundity using Tj-Gal4 driver.” - These two sentences were added to the text.

- “line 110: font size to adjust”

Answer. Font size was formatted.

- “line 210: font size to adjust”

Answer. Font size was formatted.

Reviewer 3 Report

The authors have shown a lot of efforts to revise the manuscript and the revised paper has been improved significantly. 

The authors have addressed most of my comments carefully and in detail by adding more materials in the revised version. However, the new title is not accurate enough, thus needs to be revised further.

As a result, I now recommend the current form can be accepted for publication after minor revision. 

1. For the current title “Effect of NAP Family CG5017 Hypomorphic Mutation on Human AHR Target Genes Expression in Drosophila Testis”, it sounds more like a project title, not a title for publication. What is the effect? Describe this as precise as possible.

One suggestion is, NAP family CG5017 pleiotropically regulate Human AHR Target Genes Expression in Drosophila Testis

Author Response

We are grateful to the reviewers for the positive assessment of our manuscript revision. Minor revisions in accordance to the reviewers’ suggestions were made. English language was edited by MDPI English Editing Service.  

1. “For the current title “Effect of NAP Family CG5017 Hypomorphic Mutation on Human AHR Target Genes Expression in Drosophila Testis”, it sounds more like a project title, not a title for publication. What is the effect? Describe this as precise as possible.

One suggestion is, NAP family CG5017 pleiotropically regulate Human AHR Target Genes Expression in Drosophila Testis”

Answer. We appreciated your suggestion. The title was improved.